# Self-Assembly Strategy for Synthesis of WO_3_@TCN Heterojunction: Efficient for Photocatalytic Degradation and Hydrogen Production via Water Splitting

**DOI:** 10.3390/molecules30020379

**Published:** 2025-01-17

**Authors:** Li Zhou, Wenjie Zhang, Zezhao Huang, Feng Hu, Peng Li, Xiaoquan Yao

**Affiliations:** Department of Applied Chemistry, College of Material Science and Technology, Nanjing University of Aeronautics and Astronautics, Nanjing 210016, China; zhouli@nuaa.edu.cn (L.Z.); 18856160996@163.com (W.Z.); huangzz@nuaa.edu.cn (Z.H.); fenghu@nuaa.edu.cn (F.H.)

**Keywords:** carbon nitride, heterojunction, tetracycline, photocatalytic degradation, hydrogen

## Abstract

Herein, a WO_3_@TCN photocatalyst was successfully synthesized using a self-assembly method, which demonstrated effectiveness in degrading organic dyestuffs and photocatalytic evolution of H_2_. The synergistic effect between WO_3_ and TCN, along with the porous structure of TCN, facilitated the formation of a heterojunction that promoted the absorption of visible light, accelerated the interfacial charge transfer, and inhibited the recombination of photogenerated electron–hole pairs. This led to excellent photocatalytic performance of 3%WO_3_@TCN in degrading TC and catalyzing H_2_ evolution from water splitting under visible-light irradiation. After modulation, the optimal 3%WO_3_@TCN exhibited a maximal degradation rate constant that was twofold higher than that of TCN alone and showed continuous H_2_ generation in the photocatalytic hydrogen evolution. Mechanistic studies revealed that •O_2_^−^ constituted the major active species for the photocatalytic degradation of tetracycline. Experimental and DFT results verified the electronic transmission direction of WO_3_@TCN heterojunction. Overall, this study facilitates the structural design of green TCN-based heterojunction photocatalysts and expands the application of TCN in the diverse photocatalytic processes. Additionally, this study offers valuable insights into strategically employing acid regulation modulation to enhance the performance of carbon nitride-based photocatalysts by altering the topography of WO_3_@TCN composite material dramatically.

## 1. Introduction

Antibiotics are widely used to treat disease and play a critical role in protecting human and animal health [1,2]. However, many antibiotics, such as tetracycline (TC), are difficult to fully metabolize in living organisms and therefore have long-lasting effects on the ecosystem, posing a major threat to human health and aquatic ecosystems [3]. To solve this problem, photocatalytic degradation of organic pollutants using photocatalysts is a very effective method that, like photocatalytic hydrogen production, is also a highly efficient way to utilize solar energy [4,5]. This technology works by employing a photocatalyst to absorb light energy, generating electrons and holes. This initiates a series of chemical reactions that result in the complete degradation of organic matter into harmless substances. Photocatalytic technology is characterized by its green approach, cost-effectiveness, and recyclability [6,7]. In this case, it is crucial for photocatalytic technology to be exploited for new efficient, environmentally friendly, and recyclable photocatalysts.

In 2009, the carbon nitride CN was first reported as a metal-free semiconductor photocatalyst for photocatalytic hydrogen production studies [8]. Due to its special features, such as simple preparation from monomers polymerization, outstanding physicochemical stability, and unique electronic structures, CN has been widely used in water treatment (removal of organic contaminants) and energy conversion (splitting water to produce H_2_ energy) [9,10,11]. Although, CN usually suffers from a narrow photo-responsive range and a rapid recombination rate of the charge carrier, leading to reduced photodynamic and photocatalytic rates. The photocatalytic performance of CN photocatalysts can be well modified by elemental doping [12], morphology modulation [13,14], defect construction [15], and heterostructure building [16]. Among them, both morphology modulation and heterojunction construction are endowed with significant advantages in improving the photocatalytic performance. Comparatively, the one-dimensional tubular structure of CN covers a large specific surface area, and the special hollow structure facilitates the longitudinal migration of photogenerated carriers [14]. In addition, modifying the framework structure of CN, particularly through semiconductor doping and the construction of heterojunctions, imparts outstanding properties to the composite material. This includes the efficient inhibition of the recombination of photo-generated carriers [16,17].

Researchers have developed and designed various sorts of heterojunction materials between CN and other suitable semiconductor materials to improve the photocatalytic efficiency of the CN [18,19]. Currently, Type-II system photocatalysts are considered as effective heterojunctions in photocatalysis due to the fact that the energy band arrangement facilitates the accumulation of photogenerated electrons and holes within different surface interface layers. It can realize the effective separation of electron–hole pairs and reduce the compounding chances of photogenerated electrons and holes on the surface of semiconducting materials, thus improving the photocatalytic efficiency of the catalyst. For example, Su Bao Lian constructed ZnO/g-C_3_N_4_ Type II heterojunctions, which exhibited good stability and could effectively degrade rhodamine B compared to pure g-C_3_N_4_ and pure ZnO via increasing the charge-separation efficiency [20].

Herein, tubular carbon nitride (TCN) porous nanotubes and WO_3_@TCN heterojunction materials were successfully synthesized. The WO_3_@TCN photocatalyst demonstrated superior photo-degradation activities for TC compared to unmodified TCN. The photocatalytic degradation of tetracycline solution test and photodecomposition of water for the hydrogen production test of the prepared WO_3_@TCN samples revealed that the photocatalytic activities of the modified WO_3_@TCN samples were all enhanced, with the best photocatalytic performance by the 3%WO_3_@TCN sample. Additionally, a possible photodegradation mechanism was elucidated through free radical trapping experiments. The photocatalytic enhancement mechanism could be summarized as follows: the high specific surface area provided more active sites for photocatalytic reactions, and the hollow tube structure and the formation of heterojunctions provided intra- and interlayer transfer channels for photogenerated carriers, which was conducive to the separation and transfer of photogenerated charges. Overall, the present work provides insights into material design as well as photocatalytic degradation of pollutants and photocatalytic hydrogen production.

## 2. Results and Discussion

### 2.1. Morphology

An obvious hollow tubular structure could be observed from the SEM and TEM images of TCN (Figure 1a,c). This was due to that the precursor melamine hydrothermally generating a hexagonal prismatic supramolecular structure of melamine–cyanuric acid (MC), and the center of the hexagonal prisms sublimed preferentially during thermal polymerization, resulting in the generation of a hollow micro-tubular structure. Considering that, the tubular structure was conducive to increasing the specific surface area of the catalyst; it promoted the adsorption and diffusion of reactants and active species, which, in turn, contributed to the enhancement of photocatalytic performance [21]. As illustrated in Figure 1b, following modification with WO_3_, the morphology of the 3%WO_3_@TCN composite retained its tubular structure, with the exception of spherical-shaped WO_3_ nanoparticles that were collected and aggregated on the surface of the material. When melamine underwent hydrothermal polymerization, a large amount of ammonia was released and dissolved in water, and WO_3_ reacted with ammonia to form ammonium tungstate ((NH_4_)_2_WO_4_). Subsequently, (NH_4_)_2_WO_4_ reacted with hydrochloric acid (HCl) in an acidic environment to form tungstic acid (H_2_WO_4_). Finally, WO_3_ was generated by high-temperature calcinations from H_2_WO_4_. The TEM image revealed that TCN and 3%WO_3_@TCN consisted of porous tubes with a tube size distribution from hundreds of nanometers to tens of micrometers (Figure 1c,e). As shown in Figure 1d, the image of 3%WO_3_@TCN presented distinct lattice fringes with a spacing of ~0.372 nm, consistent with the facet of WO_3_ [22]. Furthermore, EDS elemental mapping images of 3%WO_3_@TCN verified the homogeneous distribution of C, N, O, and W elements in the nanotubes, and the mass fractions of C, N, W, and O in the sample were 41.9%, 49.1%, 5.3%, and 3.7%, respectively (Figure 1f and Appendix A). The above results indicate that WO_3_ can be effectively loaded onto a carbon nitride substrate by the pH modulation method to form WO_3_@TCN heterojunctions. The formation of heterojunctions is conducive to increasing the specific surface area of the catalyst, increasing the number of active sites, and inhibiting the complexation of photogenerated electrons and holes, thus improving the photocatalytic activity.

### 2.2. Surface Chemical Composition Analysis

The surface chemical compositions and chemical bonds of TCN and 3%WO_3_@TCN were further analyzed using X-ray photoelectron spectroscopy (XPS). The XPS survey scan (Figure 2a) confirmed that both samples contained the C, N, and O elements. In the 3%WO_3_@TCN framework, two weak W 4f peaks at 35.1 and 37.0 eV demonstrated the successful WO_3_ [23] doping (Figure 2b). As shown in Appendix A, the O and W content occupied a certain mass ratio in 3%WO_3_@TCN, indicating the successful synthesis of WO_3_ in TCN. As depicted in the C 1 s spectrum of TCN (Figure 2c), the characteristic peaks at 284.8, 285.6, 288.2, 289.7, and 293.7 eV were attributed to the sp^2^ hybridized carbon atoms of C-C, C=O, C-N-C, C=NH, and N-C=N [24,25], respectively. 3%WO_3_@TCN exhibited similar peaks to TCN in the C 1s XPS spectra. The N 1s spectrum of TCN could be divided into four peaks of 398.6, 400, 401.1, and 404.3 eV (Figure 2d), attributed to C-N=C, N-(C)_3_, C-N-H, and N-O [24], respectively. The peak in the N-(C)_3_ spectrum of 3%WO_3_@TCN exhibited a noticeable shift ~0.20 eV towards a lower binding energy compared to the N 1s peak in TCN. This shift signified a weakened electron shielding effect, attributed to the increase in electron density [26].

### 2.3. Structural Investigation

The FT-IR spectrograms showed the functional groups in TCN and 3%WO_3_@TCN (Figure 3a). Given the N-H and O-H stretching of TCN-based materials, all the materials presented broad absorption bands at 3000–3400 cm^−1^. Moreover, a series of characteristic peaks at 1200–1700 cm^−1^ could be assigned to the characteristic stretching vibration peaks of C-C and C-N heterocycles [27]. The peak at 800 cm^−1^ was associated with the vibration peaks of the triazine units, which predicted the successful synthesis of the triazine ring [17]. The 3%WO_3_@TCN composite displayed a similar absorption structure to that of the pure TCN. FT-IR spectral data were consistent with those of the powder XRD analysis (Figure 3b). The XRD spectrum displayed two characteristic diffraction peaks at 13.2° and 27.4° on TCN and 3%WO_3_@TCN, respectively, which were attributed to (100) and (002) crystal planes of graphitic phase carbon nitride and the interlayer stacking formed by the 3-s-triazine ring structure and the aromatic structure [28,29]. The intensity of the characteristic diffraction peak at 27.4° was obviously reduced in 3%WO_3_@TCN, indicating that the order and crystallinity of the lamellar stacking structure of the samples were weakened, which may have been due to the dissolution and reprecipitation process of tungsten oxide affecting the formation of the C-N bond in the g-C_3_N_4_ skeleton, resulting in a weakening of the degree of interlayer stacking. However, the characteristic peaks of WO_3_ were not observed in the 3%WO_3_@TCN structure, probably due to the lower content of WO_3_.

The specific surface area and pore structure of the samples were further characterized by N_2_ adsorption–desorption isotherms (Figure 3c). The N_2_ adsorption isotherms of the TCN and 3%WO_3_@TCN materials exhibited type-IV adsorption curves, indicating the mesoporous structures of both of them [30]. The BET-specific surface areas of WO_3_, TCN, and 3% WO_3_@TCN were 10.16, 19.39, and 26.51 m^2^·g^−1^, respectively. Benefiting from the formation of porous nanotubes and the composite of two materials, 3%WO_3_@TCN exhibited a 1.4-fold higher surface area compared to TCN. Meanwhile, the pore size characteristics of the samples were calculated using the Barrett–Joyner–Halenda (BJH) model (Figure 3d). The pore size distribution exhibited similarity among the three materials in the range of 2~10 nm. However, a significant pore volume was observed in 3%WO_3_@TCN at ~2–10 nm (Appendix A), suggesting that the formation of a porous microstructure in 3%WO_3_@TCN might contribute to the generation of small, slit-like mesopores. Generally, a larger specific surface area facilitates the enrichment of material molecules at the interface, while a larger pore volume contributes to enhanced molecular diffusion efficiency. Collectively, these factors collectively improve the efficiency of the photocatalytic reaction [31]. The 3%WO_3_@TCN composites, which possess a large specific surface area and pore volume, could provide more active sites and thus accelerate the catalytic reaction efficiency.

### 2.4. Photophysical Properties

UV−vis diffuse reflection spectra were applied to research the photophysical properties and band gaps (*E*_g_) of samples. The light response of pure WO_3_ nanoparticle was mainly centered at 300 nm, while the optical absorption at wavelengths of <420 nm for TCN and 3%WO_3_@TCN was ascribed to π-π* transitions [17] in heptazine rings (Figure 4a). In generally, the high absorptive capacity of the UV spectrum facilitated an increase in the number of photogenerated electrons and photocatalytic degradation. The band gap energies of semiconductors could be estimated by the Kubelka–Munk method [23], as outlined below:*αhυ* = A(*hυ* − *E*_g_)*^n^*^/2^(1)
where *α*, *hv*, A, and *E*_g_ are the absorption coefficient, optical energy, characteristic constant of the material, and band gap energy, respectively. The value of *n* depends on the characteristics of the transition in a semiconductor. The value of *n* = 4 for the indirect semiconductors WO_3_ and TCN [23,32], so the band gap energies (*E*_g_) of WO_3_, TCN, and 3%WO_3_@TCN were estimated to be 2.65 eV, 2.78, and 2.73 eV, respectively (Figure 4b). To this end, it was hereby hypothesized that compositing two materials, WO_3_ and TCN, could reduce the band gap and effectively improve the absorption and utilization capacity of visible light, as it could take full advantage of visible light in solar energy. In order to investigate whether the potentials of VB or CB could undergo redox reactions, the valence bands (*E*_VB_) of the samples were confirmed by VB-XPS. As shown in Figure 4c, the *E*_VB_ values of WO_3_, TCN, and 3%WO_3_@TCN were approximately equal to +2.63, +2.38 V and +2.15 eV, respectively. Based on the UV–vis spectrum and VB-XPS spectra results, the conduction bands (*E*_CB_) of above samples were thus calculated to be −0.02, −0.40, and −0.58 eV, respectively. Apparently, the conduction bands of TCN and 3%WO_3_@TCN were more negative than those of (H^+^/H_2_) (*E*_0_ = 0 V, pH = 0), satisfying the thermodynamic conditions for photocatalytic decomposition of water to hydrogen. In addition, compared with TCN, the *E*_CB_ of 3%WO_3_@TCN exhibited a more negative band level, demonstrating considerable efficiency in capturing photogenerated electrons and promoting the charge separation [15]. The energy band levels of WO_3_, TCN and 3%WO_3_@TCN are schematically depicted in Figure 4d. Herein, according to the band gap values, the corresponding energy band arrangements of WO_3_ and TCN underwent an electron jump, and the electrons tended to be transferred from TCN to WO_3_. As a result, the photogenerated electrons and holes could be separated and transferred, and the complexation of the photogenerated electrons and holes was suppressed.

### 2.5. DFT Calculation and Mechanism Analysis

In order to further investigate the electron transfer path in the composite photocatalyst, the work function (Φ) values of monomers and composite photocatalysts were calculated by density functional theory (DFT). As shown in Figure 5a–c, the Φ values of TCN, WO_3_, and WO_3_@TCN were 4.66, 5.94, and 5.72 eV, respectively. The different work functions indicate the electronic transmission direction at the interface between WO_3_ and TCN. Since TCN had lower work functions, the surface charge transferred from g-C_3_N_4_ to WO_3_, which was consistent with the energy band structure. And it complied with the electron transfer path of the Type-II heterojunction [33,34]. The heterojunction structure not only improved the contact area between the photocatalysts, but also provided more active sites as well as electron transport channels, accelerating the separation of photogenerated carriers for their rapid migration to the surface of the photocatalyst. Therefore, electrons were more easily transferred from the composite photocatalyst to the surface to participate in the photocatalytic reaction in the WO_3_@TCN heterojunction. Figure 5d–f shows the simulated electron cloud structures of monomer TCN, WO_3_, and composite photocatalyst WO_3_@TCN. The simulation results show that, in the composite photocatalyst, electrons were mainly enriched at the two-phase interface of the heterojunction photocatalyst, and electrons were more inclined to be transferred from TCN to WO_3_. WO_3_ acted as an electron trap in the composite photocatalyst to capture the electrons migrating from TCN for the photocatalytic hydrogen production reaction. Meanwhile, the holes on the TCN surface were more involved in oxidation reactions, such as the degradation of pollutants.

Moreover, steady-state photoluminescence (PL) emission spectroscopy was utilized to assess the separation efficiency and recombination rates of e^−^−*h*^+^, essential factors influencing photocatalytic degradation efficiency. The PL emission spectra of the as-synthesized catalysts were excited at ~370 nm (Appendix A). It is evident that the diminished PL emission intensity of 3%WO_3_@TCN signified a reduction in the recombination rates of photo-produced charges in comparison to TCN. This resulted in an enhanced charge separation, an extended lifetime, and a proliferation of intimate interfaces between WO_3_ NPs and TCN nanotubes [35]. This phenomenon could be attributed to the heterojunction formed in the contact layer between WO_3_ and TCN, which is likely to have reduced the recombination probability of photogenerated electron–hole pairs. Heterojunction catalysts proved to be effective in reducing carrier recombination, facilitating the separation and transfer of photogenerated charges. This enhancement in the transfer and separation of photo-excitation-generated carriers is highly beneficial for achieving optimal photocatalytic performance.

### 2.6. Photocatalytic Degradation of Tetracycline

In this study, the photocatalytic activity of the catalysts was evaluated using TC as the degradation prototype. Prior to illumination, a 40 min adsorption–desorption equilibrium dark reaction was performed. As shown in Figure 6a, the concentrations of TC remained largely unaltered in the absence of the catalyst. However, the introduction of different types of catalysts could have resulted in varying degrees of degradation of pollutants. As a control, the degradation rates of TC by WO_3_ and TCN were 10.6% and 67.3%, respectively, within 80 min. These results indicated that WO_3_ did not possess significant photocatalytic properties, while TCN exhibited higher degradation efficiency. Notably, the combination of WO_3_ with TCN resulted in the WO_3_@TCN composite exhibiting enhanced degradation efficiency compared to the individual materials. We also conducted three parallel experiments, and the 3%WO_3_@TCN photocatalyst showed stable photocatalytic degradation efficiency of tetracycline (Appendix A).

The apparent rate constant *k* of the photocatalytic samples was analyzed, and the values of WO_3_, TCN, 1%WO_3_@TCN, 3%WO_3_@TCN, and 5%WO_3_@TCN were calculated to be 0.000, 0.012, 0.024, 0.025, and 0.017 min^−1^, respectively (Figure 6b). Among all the samples, 3%WO_3_@TCN presented the maximal degradation rate constant, which was twofold that of TCN. However, given that the excessive tungsten oxide covered the active site, the excessive amount of WO_3_ dispersed on the surface of TCN led to a decrease in photocatalytic efficiency. The 3%WO_3_@TCN heterojunction photocatalyst, prepared in this experiment, exhibited an optimal degradation rate even when a minimal addition was utilized.

In order to emphasize the photocatalytic degradation performance of 3%WO_3_@TCN, we compared the catalytic degradation ability of 3%WO_3_@TCN with other modified g-C_3_N_4_ samples reported in the literature for TC degradation [36,37,38,39,40]. As can be seen from Appendix A, under the same light conditions, the 3%WO_3_@TCN catalyst exhibited superior photocatalytic degradation efficiency for TC, with a lower catalyst dosage and shorter degradation time. Therefore, the photocatalytic performance of g-C_3_N_4_ can be effectively improved by constructing heterojunction, which provides a certain reference value for the structural modification of the material. The stability of 3%WO_3_@TCN was evaluated by three consecutive photocatalytic degradation tests, as shown in Appendix A. The degradation efficiency of 3%WO_3_@TCN for tetracycline remained 76.3% after three cycles, indicating that the 3%WO_3_@TCN samples had stable photocatalytic degradation activity.

To further understand the details of degradation, isopropanol (IPA), L-ascorbic acid (LAA), and triethanolamine (TEOA) served as the capturing agents of the •OH, •O_2_^−^, and *h*^+^ species [6,32,41], respectively. In the free radical capture experiment involving the photocatalytic degradation of TC by 3%WO_3_@TCN, the additions of IPA, LAA, and TEOA each had distinct effects on the removal rate of TC to varying degrees (Figure 6c,d). Among them, the TC removal rate decreased most obviously when LAA was the quencher, demonstrating •O_2_^−^ as the main ROS participating in the reaction, whereas the degradation of TC was only mildly inhibited in the presence of IPA and TEOA. Notably, despite the fact that *h*^+^ was quenched in the presence of TEOA, the removal rate of TC was only reduced by ca. 10%, suggesting that the oxidation of TC by *h*^+^ was not the main pathway in this photocatalytic degradation reaction. In addition, the DMPO spin trapping technique also verified the generation of •O_2_^−^ in the photocatalytic reaction (Appendix A). There was no ESR signal peaks when the catalyst was not photo-exposed, whereas the ESR signal peaks of •O_2_^−^ could be clearly observed in the reaction system after the catalyst was treated with photo-exposure for 1 min, thus further verifying the generation of •O_2_^−^ under photo-exposure conditions by 3%WO_3_@TCN radicals as the main active species in the degradation process.

Based on the above mechanism investigation, a plausible photocatalytic oxidation mechanism was suggested for the 3%WO_3_@TCN composite (Figure 7). For unmodified TCN, the rapid electron–hole recombination limited the photocatalytic property. In contrast, the band gap of 3%WO_3_@TCN (2.73 eV) was narrower than that of TCN (2.78 eV) (Figure 4b), and more photogenerated carriers could be generated under visible light irradiation. Under visible light irradiation, photo-excited electron–hole pairs were generated and migrated to the surface of 3%WO_3_@TCN heterostructure. Exploiting the staggered energy bands between WO_3_ and TCN, e^−^ transferred from TCN to the CB of WO_3_ and was trapped by O_2_ to form superoxide radical •O_2_^−^. Meanwhile, the *h*^+^ was transferred from WO_3_ to the VB band of TCN. Finally, reactive oxygen species (•O_2_^−^) effectively acted with TC, leading to its degradation. All in all, the formation of a heterojunction effectively inhibited the recombination of photogenerated electron–hole pairs and thus improved the photocatalytic activity.

### 2.7. Visible Photocatalytic Hydrogen Production Performance

The photocatalytic activity of TCN and 3%WO_3_@TCN was also evaluated via hydrogen evolution from water splitting with methanol as a sacrificial agent. As shown in Appendix A, both TCN and 3%WO_3_@TCN showed continuous H_2_ generation on the samples, indicating their photocatalytic activity in hydrogen production. Notably, in the first hour, the 3%WO_3_@TCN sample presented a higher H_2_ generation efficiency than that of the unmodified TCN. Then, the hydrogen production of the 3%WO_3_@TCN displayed a slightly decrease, which may have been due to the unexpected aggregation of catalysts and/or the light shielding effect caused by the loading amount of WO_3_ [42,43].

## 3. Experimental Section

### 3.1. Chemicals and Materials

Melamine (MA, C_3_H_6_N_6_, 99%), tungstic Oxide (WO_3_, 99%), hydrochloric acid (HCl, 37%), methanol (CH_3_OH, 99%), and ethyl alcohol (C_2_H_5_OH, 99%) were provided by Beijing Inno Chem Science & Technology Co., Ltd. (Beijing, China). Tetracycline (C_22_H_24_N_2_O_8_, abbreviated as TC, 98%), and chloroplatinic acid hydrate (H_2_PtCl_6_·6H_2_O, 99.9%) was purchased from Aladdin (Bay City, MI, USA). Triethanolamine (C_6_H_15_NO_3_, 99%), L-Ascorbic acid (C_6_H_8_O_6_, 99%), and isopropanol (C_3_H_8_O, 99%) were purchased from Macklin Co., Ltd. (Shanghai, China). Other chemical reagents involved were analytically pure without purification. Meanwhile, deionized water (>18.2 MΩ•cm) was obtained from a Milli-pore Milli-Q system (High-tech, Shanghai, China) used throughout the experiments.

All instruments and characterizations adopted in this experiment are detailed in the Appendix A section.

### 3.2. Synthesis of Carbon Nitride Nanotubes (TCN)

Following a slightly modified method from the literature [44], 3.0 g of melamine (MA) was dispersed into 60 mL of ultrapure water and stirred at 90 °C for 24 h. The melamine solution was then transferred to a 100 mL Teflon-lined stainless steel autoclave and hydrothermally processed at 180 °C for 24 h. Subsequently, the precursor product was isolated by centrifugation, washed with ultrapure water and ethanol several times, and dried in an oven at 80 °C overnight. The above product was placed into an alumina crucible and heated from 30 °C to 500 °C at a 2 °C/min ramping rate under an N_2_ atmosphere. Then, the sample was naturally cooled to 30 °C, and a yellow powder was yielded containing carbon nitride nanotubes (TCNs).

### 3.3. Synthesis of WO_3_ Modified WO_3_@TCN Nanorods (WO_3_@TCN)

Typically, WO_3_ (30.0 mg) was added to the melamine solution and stirred for another 30 min. Following that, the solution was transferred into a 100 mL Teflon-lined stainless steel autoclave and subjected to hydrothermal processing at 180 °C for 24 h. The solution was then allowed to cool naturally to room temperature, and the pH was adjusted to 5~6 using hydrochloric acid. The precursor product was isolated by centrifugation, thoroughly washed with ultrapure water and ethanol multiple times, and finally dried overnight. The solid was dried in an oven at 80 °C overnight to obtain the modified precursor product, which was placed into an alumina crucible and heated from 30 °C to 500 °C at a 2 °C/min ramping rate under an N_2_ atmosphere. Then, the sample was naturally cooled to 30 °C, and a powder was yielded with WO_3_-modified carbon nitride nanotubes (1%WO_3_@TCN). Finally, 3% and 5%WO_3_@TCN were also prepared according to the above procedure. Figure 1 presents the preparation process diagram.

### 3.4. Measurements of Photocatalytic Degradation Performance

To facilitate the photodegradation of TC under solar irradiation (using a 300 W Xeon lamp, Beijing Magnesium Ruichen Technology Co., Ltd., Beijing, China) at room temperature, 10 mg of the catalyst was dispersed in a 50 mL of TC solution (20 mg/L). The mixture was sonicated for 40 min to reach equilibrium under dark conditions. During irradiation, the temperature of the reaction solution was kept at 25 °C by passing circulating water through the reaction tube. Furthermore, the suspension was filtered by membrane filtration (pore size: 0.45 µm) and detected using a UV-vis spectrophotometer (UV-2500, Shimadzu Corporation, Tokyo, Japan). The actual power density was 59.8 mW/cm^2^ for the light irradiation in the photocatalytic experiments.

For the free radical quenching experiments, 50 mL of TC solution was supplemented with 5 mM triethanolamine (TEOA), 5 mM isopropanol (IPA), and 5 mM L-Ascorbic (LAA) to quench *h*^+^, •OH, and •O_2_^−^, respectively [41,45].

### 3.5. Photocatalytic Hydrogen Evolution from Water Splitting on WO_3_@TCN with Pt as the Cocatalyst

In the photodeposition process of Pt (0.5 wt %) on the sample, 40 mg of the sample was dispersed through ultrasonication in a methyl alcohol aqueous solution (V_water_:V_methanol_ = 55 mL:15 mL) within a reaction cell. Subsequently, a solution of H_2_PtCl_6_·6H_2_O at a concentration of 0.5 M was added to the reaction cell. The system was then connected to the evaluation system. The evaluation of photocatalytic overall water-splitting performance was conducted in a closed gas circulation system equipped with an overhead-irradiation-type glass vessel. Prior to each test, all the air was removed and 1 kPa Ar was injected to facilitate the detection of little generated gases. A 300 W Xe lamp was adopted as the irradiation source (equipped with a 420 nm cut-off filter). During each reaction, the suspension was kept at 25 °C, and the evolved gases were analyzed by gas chromatography (GC-2014c).

## 4. Conclusions

In conclusion, the WO_3_@TCN photocatalyst was successfully synthesized using a self-assembly method and demonstrated effectiveness in the degradation of organic dyestuffs and the photocatalytic evolution of H_2_. In addition, leveraging the porous structure and high specific surface area of TCN, the formation of a heterojunction promoted the absorption of visible light, accelerated the interfacial charge transfer, and inhibited the recombination of photogenerated electron–hole pairs, which contributed to the excellent photocatalytic performance of 3%WO_3_@TCN in the degradation of TC under visible-light irradiation. DFT theoretical calculations verified that the electrons were favored to migrate to the surface of the WO_3_@TCN heterojunction and participated in the photocatalytic reaction, thus promoting the adsorption and degradation of pollutants on the catalyst surface. Overall, this study provides a new perspective for the structural design of green TCN-based heterojunction photocatalysts and expands the application of TCN in diverse photocatalytic applications.

## Data Availability

The original contributions presented in this study are included in the article/Appendix A. Further inquiries can be directed to the corresponding authors.

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
