# Peer review of "Self-Assembly Strategy for Synthesis of WO3@TCN Heterojunction: Efficient for Photocatalytic Degradation and Hydrogen Production via Water Splitting"

_molecules, 2025, doi:10.3390/molecules30020379_

Round 1
Reviewer 1 Report
Comments and Suggestions for Authors
The article addresses the synthesis and characterization of a heterojunction, as well as the applications of this material in the degradation of an antibiotic and hydrogen production. The procedures used for material synthesis are clearly and comprehensively described. The characterizations conducted are thorough, covering different relevant aspects of the material. The results obtained demonstrate promising performance in the degradation of the antibiotic and in hydrogen production. However, for the work to be accepted, some issues need to be clarified, and specific recommendations must be followed.
- Is the material obtained novel?
- Is the synthetic route used to obtain the WO3@TCN heterojunction original? If not, cite the article that describes this route. Additionally, justify the choice of this route over others available in the literature.
- What new contributions does this material bring compared to others described in the literature? It is essential to include a section in the results and discussion detailing these aspects, highlighting the innovations of the proposed material.
Considerations
a) The results related to tetracycline degradation and hydrogen production should be compared with other data reported in the literature. It is important to detail the advantages and disadvantages of the material described in this work compared to existing ones.
Author Response
- Is the material obtained novel?
We sincerely thank the referee’s comments,our material is a new WO3@TCN heterojunction. The literature has reported WO3/ g-C3N4 graphitic or WO3/g-C3N4 nanotubes, however, the carbon nitride we synthesized is microtubular structure. The method and structure are different from previous in WO3/g-C3N4 heterojunction.
- Is the synthetic route used to obtain the WO3@TCN heterojunction original? If not, cite the article that describes this route. Additionally, justify the choice of this route over others available in the literature.
We sincerely thank the referee’s comments. The TCN was synthesized with slight modifications according to the method of the reference (Nano Research, 2018, 11(6): 3462–3468 ), and we have cited relevant literature in the article. WO3@TCN is a new synthesis method and the formation mechanism has been described in detail in the morphology section.
- What new contributions does this material bring compared to others described in the literature? It is essential to include a section in the results and discussion detailing these aspects, highlighting the innovations of the proposed material.
We sincerely thank the referee’s comments, since we did not express it clearly. Tetracycline has caused significant residues in the environment due to its extensive use in livestock and aquaculture, and has become a prominent environmental risk issue. Typically, tetracycline is more difficult to degrade; nevertheless, WO3@TCN exhibits excellent photocatalytic degradation of tetracycline. We compared the synthesized material with other carbon nitride in tetracycline degradation work. An additional supplement comparative data are provided in Table. S3. In the original article, we also added a comparison of the photocatalytic properties of different materials.
The additions to the original text are as follows:
In order to emphasize the photocatalytic degradation performance of 3%WO3@TCN, we compared the catalytic degradation ability of 3%WO3@TCN with other modified g-C3N4 samples for TC degradation reported in the literature. As can be seen from Table S3, under the same light conditions, the 3%WO3@TCN catalyst exhibited superior photocatalytic degradation efficiency for TC, with less catalyst dosage and shorter degradation time. Therefore, the photocatalytic performance of g-C3N4 can be effectively improved by constructing heterojunction, which provides a certain reference value for the structural modification of the material.
Reviewer 2 Report
Comments and Suggestions for Authors
In this paper (molecules-3419957), titled "Self-assembly strategy for synthesis of WO3@TCN heterojunction: Efficient for photocatalytic degradation and hydrogen production via water splitting," the authors presented a WO3@TCN heterojunction photocatalyst from self-assembly strategy, demonstrating effectiveness of photocatalytic degradation of pollutants and H2 evolution. Experimental and DFT calculations verified promoted visible light absorption and enhanced charge carriers separation, thus leading to favorable photocatalytic efficiency. This work is interesting, and the manuscript can be suitable for potential publication in Molecules after addressing the following comments:
1. Since the heterojunction concept is crucial to this work, it is recommended to expand the discussion in Section 1 to include the different types of heterojunctions commonly studied in photocatalytic applications.
2. The actual power density for the light irradiation in the photocatalytic experiments must be determined (W/cm2) and stated in the manuscript.
- Is CN a direct bandgap semiconductor or an indirect bandgap semiconductor (Science, 2015, 347(6225): 970-974)? The authors might regulate the band structures of TCN and 3%WO3@TCN (Fig. 4b).
4. In order to demonstrate the reproducibility of the performance, it should repeat the photocatalytic measurements (Fig. 6a) to obtain the error bars.
5. What is the concentration of the scavengers? There is controversy over the use of isopropanol as hydroxyl radical scavenger, as isopropanol can also react with holes. It is better to use the other scavenger, such as tert-butanol.
- As an important capacity of catalyst, the reusability of the photocatalyst should be tested. What is the stability of 3%WO3@TCN for photocatalytic degradation of TC after several rounds of the cycling process?
- The authors stated that the calculated results align with the theoretical calculations from Yu’s group (https://doi.org/10.1016/j.apcatb.2018.11.011). However, the direction of electron transfer appears to differ. Please double-check and clarify the type of heterojunction between WO3 and TCN (e.g., type-II or S-scheme heterojunction), and provide additional evidence to support the conclusion.
Author Response
- Since the heterojunction concept is crucial to this work, it is recommended to expand the discussion in Section 1 to include the different types of heterojunctions commonly studied in photocatalytic applications.
We sincerely thank the referee’s comments, your suggestion is good. We have added an exploration of the different types of heterogeneous nodes in Section 1.
The article is below: Researchers have developed and designed various sorts of heterojunction materials between CN and other suitable semiconductors materials to improve the photocatalytic efficiency of the CN[18-19]. Currently, Type-II system photocatalysts are considered as effectively heterojunction in photocatalysis, due to that the energy band arrangement facilitates the accumulation of photogenerated electrons and holes within different surface interface layers. It can realize the effective separation of electron-hole pairs and reduce the compounding chances of photogenerated electrons and holes on the surface of semiconducting materials, thus, improved the photocatalytic efficiency of the catalyst. For example, Su Bao Lian constructed ZnO/g-C3N4 Type II heterojunction, which exhibit good stability and can effectively degrade rhodamine B comparing to pure g-C3N4 and pure ZnO via increasing the charge-separation efficiency[20].
- The actual power density for the light irradiation in the photocatalytic experiments must be determined (W/cm2) and stated in the manuscript.
We sincerely thank the referee’s comments. We measured the optical density of the xenon lamp light source at different locations using a five-point method with an optical power meter, and the results have added to “Measurements of photocatalytic degradation performance”.
The actual power density is 59.8 mW/cm2 for the light irradiation in the photocatalytic experiments.
- Is CN a direct bandgap semiconductor or an indirect bandgap semiconductor (Science, 2015, 347(6225): 970-974)? The authors might regulate the band structures of TCN and 3%WO3@TCN (Fig. 4b).
We sincerely thank you for your insightful comments and suggestions. Carbon nitride can exhibit either direct semiconductor properties or indirect semiconductor properties due to its different internal structures. Both experimental and theoretical computational studies had demonstrated that the forbidden band width range of carbon nitride is about 2.5 ~ 2.8 eV. We found that the band gap value (Eg = 2.78 eV) of carbon nitride obtained by fitting the Kubelka-Munk method equation using n = 4, which showed a good linear fit and accord with previous literature results (Applied Catalysis B: Environment and Energy, 2024(357):124301), claiming TCN to be an indirect band gap material. The figure.1 shows the band gap diagram of carbon nitride obtained by fitting using n = 1. The band gap value of carbon nitride in literature is also obtained by fitting the Kubelka-Munk equation (Science, 2015, 347(6225): 970-974).
Fig 1. Kubelka-Munk equation fitting plot for n=1of 3%WO3@TCN.
- In order to demonstrate the reproducibility of the performance, it should repeat the photocatalytic measurements (Fig. 6a) to obtain the error bars.
We sincerely thank the referee’s comments. A repeated experiment on the photocatalytic degradation of tetracycline by 3%WO3@TCN has been added in Fig. S3. Parallel experiments showed that 3%WO3@TCN had a consistent degradation trend each time in the photocatalytic degradation of tetracycline.
- What is the concentration of the scavengers? There is controversy over the use of isopropanol as hydroxyl radical scavenger, as isopropanol can also react with holes. It is better to use the other scavenger, such as tert-butanol.
We sincerely thank the referee’s comments, since we did not put this section in a conspicuous position, we are sorry for your misunderstanding. The concentration of the scavengers has described in Experimental Section 2.4. For the free radical quenching experiments, 50 mL of TC solution was supplemented with 5 mM triethanolamine (TEOA), 5 mM isopropanol (IPA), and 5 mM L-Ascorbic (LAA) to quench h +, •OH and •O2-, respectively.
Both isopropanol and tert-butanol can be hydroxyl radical trapping agents. The following papers all used isopropanol as a hydroxyl radical trapping agent (Acs Applied Materials & Interfaces, 2024, 16: 784-794; International Journal of Hydrogen Energy, 2024, 86: 1326-1336; Applied Catalysis B: Environmental, 2017, 206: 417-425). At the same time, we feel that the scope of the current thesis can still support the argument of this article. Therefore, we recommend that the supplementary experiment be included in follow-up work in the future.
- As an important capacity of catalyst, the reusability of the photocatalyst should be tested. What is the stability of 3%WO3@TCN for photocatalytic degradation of TC after several rounds of the cycling process?
We sincerely thank the referee’s comments. The photocatalytic cycle experiment to degrade tetracycline by 3%WO3@TCN has been added in Fig. S4. The cyclic photocatalytic degradation activity of TC did not decrease significantly, indicating that the 3%WO3@TCN samples have stable photocatalytic degradation activity.
- The authors stated that the calculated results align with the theoretical calculations from Yu’s group (https://doi.org/10.1016/j.apcatb.2018.11.011). However, the direction of electron transfer appears to differ. Please double-check and clarify the type of heterojunction between WO3and TCN (e.g., type-II or S-scheme heterojunction), and provide additional evidence to support the conclusion.
We sincerely thank the referee’s comments, we are sorry for our careless mistake. We agree that more in-depth study should be conformed to details of type-II or S-scheme heterojunction. The work functions of DFT and band gap have confirmed the surface charge transfers direction of WO3@TCN heterojunction in accordance with type-II heterojunction. Some literatures have shown that the heterojunction material transforms from type II to S-scheme in some specific reaction conditions (Nature communincations, 2023, 14: 3901; Journal of Colloid and Interface Science, 2024, 675: 150-191; Separation and Purification Technology, 2024, 345: 127323). Consequently, we believe that the inner electrons transfer under Xe light need to detect by In-situ XPS photoelectron spectroscopy. However, due to limitations such as laboratory conditions and lack of appropriate experiments tools, it is impractical to conduct the related experiment at present. Based on the results of previous studies, we concluded that electron transport encompasses type-II heterojunction and cited relevant literature to support our paper.
Round 2
Reviewer 2 Report
Comments and Suggestions for Authors
The revised manuscript can be accepted for publication since the authors have made reasonable responses and revision.